# Stage-Dependent Fibrotic Gene Profiling of WISP1-Mediated Fibrogenesis in Human Fibroblasts

**DOI:** 10.3390/cells13232005

**Published:** 2024-12-05

**Authors:** Kirti Singh, Marta Witek, Jaladhi Brahmbhatt, Jacquelyn McEntire, Kannan Thirunavukkarasu, Sunday S. Oladipupo

**Affiliations:** 1Biotherapeutics Enabling Biology, Lilly Research Laboratories, Eli Lilly and Company, Indianapolis, IN 46225, USA; kirti.singh481@gmail.com (K.S.); mcentire_jacquelyn@lilly.com (J.M.); 2Protein Optimization, Lilly Research Laboratories, Eli Lilly and Company, Indianapolis, IN 46225, USA; marta.witek@lilly.com (M.W.); brahmbhatt_jaladhi@lilly.com (J.B.); 3Immunology Research, Lilly Research Laboratories, Eli Lilly and Company, Indianapolis, IN 46225, USA; thirunavukkarasu_kannan@lilly.com

**Keywords:** WISP1/CCN4, idiopathic pulmonary fibrosis, fibroblast, inflammation, extracellular matrix, matricellular protein

## Abstract

Idiopathic pulmonary fibrosis (IPF) is the most common interstitial lung disease with unknown etiology, characterized by chronic inflammation and tissue scarring. Although, Pirfenidone and Nintedanib slow the disease progression, no currently available drugs or therapeutic interventions address the underlying cause, highlighting the unmet medical need. A matricellular protein, Wnt-1-induced secreted protein 1 (WISP1), also referred to as CCN4 (cellular communication network factor 4), is a secreted multi-modular protein implicated in multi-organ fibrosis. Although the precise mechanism of WISP1-mediated fibrosis remains unclear, emerging evidence indicates that WISP1 is profibrotic in nature. While WISP1-targeting therapy is applied in the clinic for fibrosis, detailed interrogation of WISP1-mediated fibrogenic molecular and biological pathways is lacking. Here, for the first time, using NanoString^®^ technology, we identified a novel WISP1-associated profibrotic gene signature and molecular pathways potentially involved in the initiation and progression of fibrosis in primary human dermal and lung fibroblasts from both healthy individuals and IPF patients. Our data demonstrate that WISP1 is upregulated in IPF-lung fibroblasts as compared to healthy control. Furthermore, our results confirm that WISP1 is downstream of the transforming growth factor-β (TGFβ), and it induces fibroblast cell proliferation. Additionally, WISP1 induced IL6 and CCL2 in fibroblasts. We also developed a novel, combined TGFβ and WISP1 in vitro system to demonstrate a role for WISP1 in the progression of fibrosis. Overall, our findings uncover not only similarities but also striking differences in the molecular profile of WISP1 in human fibroblasts, both during the initiation and progression phases, as well as in disease-specific context.

## 1. Introduction

Fibrosis is a highly progressive, irreversible, and chronic inflammatory condition characterized by excessive extracellular matrix (ECM) deposition, tissue architectural remodeling, and scarring. Fibroblasts are majorly responsible for synthesizing and organizing ECM proteins, playing a central role in wound healing and repair. However, persistent injury and chronic insult dysregulates and disrupts the normal physiological restorative wound-healing process, tipping the balance toward pathophysiological conditions [1]. A cellular mechanism, including but not limited to fibroblast proliferation, fibroblast to myofibroblast transition (FMT), epithelial injury, epithelial to myofibroblast transition (EMT), and apoptosis-resistant fibroblast phenotype, is central to fibrosis pathology associated with scleroderma (skin), idiopathic pulmonary fibrosis (IPF) (lung), cardiac fibrosis (heart), renal fibrosis (kidney), and liver fibrosis, amongst other organs [2]. Currently, clinical manifestations of the disease are managed by two Food and Drug Administration (FDA)-approved small molecule inhibitors, Nintedanib and Pirfenidone, which slow the disease progression [3]. However, both drugs are associated with gastrointestinal side-effects, such as vomiting, diarrhea, abdominal pain, appetite loss, and weight loss. Although these adverse effects are manageable, they significantly affect patient compliance and overall quality of life [4]. Furthermore, the median survival rate of IPF patients after diagnosis is approximately 3–5 years, underscoring the urgent need to identify novel and promising therapeutic targets for the treatment of fibrosis.

WISP1 is a highly conserved, secreted cysteine-rich, multi-modular matricellular protein. Emerging evidence suggests that WISP1 is profibrotic in nature and drives a myriad of cellular processes, including cell proliferation, migration, and invasion. Large-molecule therapeutic modalities, such as WISP1-neutralizing antibody, have been shown to elicit protective effects, as we and others have previously reviewed [5,6]. Furthermore, a recent study has identified a novel role for WISP1 in driving the progression of liver fibrosis. WISP1 knockout animals demonstrated protection in a CCl_4_-liver fibrosis model, highlighting WISP1 as a promising therapeutic target for drug development. Besides liver fibrosis, aberrantly increased WISP1 expression in both preclinical animal models of pulmonary fibrosis (paraquat-induced model, bleomycin model), and clinically, in IPF-lung as compared to healthy control, has been positively correlated with the pathophysiology of fibrosis. In fact, first-in-human phase-I clinical trials are underway to assess the safety, tolerability, and pharmacokinetics of the anti-WISP1 antibody (MTX-463) (NCT06401213). Although recent mechanistic studies have enhanced our understanding of the molecular mechanisms involved in WISP1-mediated liver fibrogenesis [7], WISP1 remains an orphan ligand since its cognate receptor has yet to be discovered, limiting further mechanistic insights. Furthermore, the key underlying molecular mechanisms and associated biological pathways of WISP1-mediated fibrogenesis in diseased vs. non-diseased lung and dermal fibroblasts still remain poorly understood.

In this study, we aim to enhance our understanding of the role of WISP1 in lung and dermal fibrosis by transcriptional profiling of WISP1 to better characterize its molecular signature in the initiation and progression of disease. By utilizing NanoString^®^-based technology, we identified unique gene networks and pathways across multiple organ-specific fibroblasts. This allowed us to better characterize the complex role of WISP1 in both diseased and non-diseased states. For the first time, we reveal a WISP1-specific gene signature and associated downstream signaling patterns for fibrosis initiation and progression in primary human healthy versus IPF-diseased as well as primary dermal fibroblasts. Our results also reveal notable differences in WISP1-mediated gene expression between healthy and IPF-diseased lung fibroblasts, as well as between lung and dermal fibroblasts. These findings highlight the heterogeneity of fibroblasts and tissue-specific differences.

## 2. Materials and Methods

### 2.1. Cell Culture

Primary human lung fibroblasts from Idiopathic Pulmonary Fibrosis-diseased (DHLF), healthy (NHLF) donor, and normal primary adult human dermal fibroblasts (NHDF) were obtained from Lonza (Cat. # CC-2512, CC-7231, CC-2511, respectively; Basel, Switzerland). Primary cells were cultured in T-75 cm^2^ flasks containing fibroblast growth basal media (FBM) (CC-3131) supplemented with insulin 0.5 mL, gentamicin sulfate 0.5 mL, human fibroblast growth factor-B (hFGF-B) 0.5 mL, and fetal bovine serum 10 mL at 37 °C in 5% CO_2_, as recommended by the manufacturer. For serum starve media, we used FBM without supplements. In addition, the same donor-specific lots were used in all the experiments (till passage 4) to ensure reliability and consistency. All the lots were commercially validated and tested positive for smooth-muscle alpha actin expression and negative for epithelial cell-marker cytokeratin 14 (CK14), cytokeratin 18 (CK18), and cytokeratin 19 (CK19) expression, ascertaining the purity of the cultured fibroblasts. Healthy NHLF and DHLF were obtained from 5 different donors, while NHDF were obtained from 3 different donors. The clinical characteristics of the patient-derived cell lines are listed in Table 1. Commercially acquired primary cells were expanded and multiple frozen aliquots at early passage were cryopreserved. Cells were passaged a maximum of 3–4 times after which another parent aliquot of the same lot was used.

### 2.2. Treatment

NHLF, DHLF, and NHDF (4 × 10^5^ cells/well) cells were seeded in a 6-well plate and allowed to adhere overnight and then serum-starved for 24 h prior to treatment with either TGFβ (1 ng/mL) or recombinant human WISP1 (1000 nM) in serum starve media for 24 h. To study progression, cells were incubated with TGFβ1 (7754-BH; R&D) (1 ng/mL) for 24 h prior to stimulation with WISP1 (1000 nM) for another 24 h. Following treatment, cell culture supernatant and RNA were collected for analysis. Recombinant human WISP1 was generated in-house. HEK293E cells were transiently transfected using polyethyleneimine hydrochloride (PEI MAX; 24765 (100)) in a 2 L shake flask at 37 °C, 5–8% CO_2_ at 160 rpm. Cells were cultured for 5 days and spun at 7000× *g* for 20 min prior to protein extraction using a 5 mL Hi trap His-excel column and a 50 mL Q-FF anion-exchange chromatography column.

### 2.3. Quantitative Real-Time PCR

Total RNA for cultured cells was extracted using a PureLink RNA mini-kit (Cat#12183018A; Invitrogen, Carlsbad, CA, USA) and quantified using a Nanodrop Eight Spectrophotometer (Thermo Scientific, Waltham, MA, USA). Complementary DNAs were synthesized by reverse transcription with 100 ng of total RNA using SuperScript III First strand (Cat#18080-051; Invitrogen, Carlsbad, CA, USA). The following commercially available TaqMan probes and primers were utilized: human Col1A1 (Hs00164004_m1), Col3A1 (Hs00943809_m1), CCL2 (Hs00234140_m1), IL6 (Hs00985639_m1), ACTA2 (Hs05005341_m1), FN1 (Hs01549976_m1), CCN4/WISP1 (Hs00987448_m1), and 18S (4310893E) from Applied Biosystems. cDNAs (1:20 dilution ratio) were added along with TaqMan Fast-advanced Mastermix in a total 10 µL reaction volume in a 384-well plate as per the manufacturer’s instructions and the plate was read using Quant Studio 7 Flex (Applied Biosystems, Waltham, MA, USA). All the genes were normalized to endogenous 18S. Gene expression was determined using the 2^ΔΔCt^ method. Briefly, ΔCt was calculated by subtracting the sample Ct value from the Ct value of the 18S reference gene. ΔΔCt was calculated by subtracting the average ΔCt value of unstimulated control from the sample ΔCt value. Relative quantification of gene expression was calculated by 2^−ΔΔCt^. Each reaction was performed in triplicates.

### 2.4. ELISA

Cell culture supernatants were collected post-treatment and stored in −20 °C until ready to analyze. Supernatants were thawed at room temperature and diluted 1:10-fold in PBS. WISP1 (DY1627), IL-6 (D6050), and CCL-2 (DCP00) protein content was assessed using an enzyme-linked immunosorbent assay (ELISA) kit from R&D systems as per the manufacturer’s recommendations. Protein standard and samples from three biological replicates were run in duplicate. Optical density was determined at 450 nm with wavelength correction set to 540 nm using a SpectraMax^®^ i3x (Molecular Devices, San Jose, CA, USA) microplate reader.

### 2.5. Cell Proliferation Assay

A total of 5 × 10^3^ cells were seeded per well in a 96-well black, clear-bottom plate in total 100 µL growth media and culture overnight. Cells were serum-starved for 96 h prior to treatment with CCN4 or PDGF-BB (positive control) for 24 h. An equal volume of 4% paraformaldehyde solution was added per well and incubated for 15 min at room temperature. Post-fixation, cells were washed three times with 1X PBS solution and blocked in 2% normal goat serum (in PBS) for 1 h at room temperature with gentle rocking. Cells were incubated with 1:1000 pRb primary antibody (Rabbit anti-phospho-Rb (Ser807/811) -D20B12) AF488 Ab used at 1:2000 dilution (Cat. #4277S; Cell Signaling Tech, Danvers, MA, USA) and Nuc Blue live cell stain (Cat. #R37605) in PBS containing 0.1% triton-X 100 overnight at 4 °C. Cells were washed three times with 1X PBS solution and images were acquired using a Phenix Opera microscope (Revvity, Waltham, MA, USA). %pRb positive cells were analyzed using Harmony high-content imaging software5.1 (Revvity, Waltham, MA, USA).

### 2.6. NanoString^®^ Gene Analysis

An nCounter Human Fibrosis V2 panel (NanoString^®^ Technologies, Inc., Seattle, WA, USA) comprised of 770 genes was utilized to assess the fibrotic gene signature. A quantity of 100 ng of total RNA isolated from cells was hybridized with a capture and reporter probe set at 65 °C for 20 h in a PCRmax Alpha Cycler 4 thermal cycler (Cole-Parmer, Vernon Hills, IL, USA) following the manufacturer’s instructions. The samples were processed on an nCounter MAX/FLEX system followed by cartridge scanning by digital analyzer. The .rcc files obtained were uploaded as the input data and normalized using housekeeping genes included in the panel using NanoString nSolver analysis software version 4.0 (nSolver 4.0, NanoString Technologies, Inc., Seattle, WA, USA). The background (thresholding count was set to 20 and pairwise ratios were generated to compare different groups. Each sample was performed at least in triplicates.

### 2.7. Data Analysis

All the statistical analysis were performed and graphed using GraphPad Prism 10 (San Diego, CA, USA) and RStudio 3.6.0 (Boston, MA, USA). Data are represented as mean ± SD and experiments were performed at least three independent times in triplicates as noted in the figure legends. Statistical analysis was performed using one-way analysis of variance (ANOVA) and Tukey’s post hoc analysis with ninety-five percent confidence interval and statistical significance represented as * *p* < 0.5, ** *p* < 0.01, *** *p* < 0.001, **** *p* < 0.0001, as described in the figure legends.

## 3. Results

### 3.1. WISP1 mRNA and Protein Are Upregulated in IPF Lung Fibroblasts

To investigate the role of WISP1 in initiating fibrosis, we first examined and compared both basal mRNA expression along with secreted WISP1 protein levels in healthy control (NHLF) and IPF-derived (DHLF) primary human lung fibroblasts obtained from 5 donors each. WISP1 was highly upregulated at both transcript (Figure 1A) and protein levels (Figure 1B) in DHLF as compared to NHLF. The results also demonstrated variability in endogenous WISP1 expression among DHLF patients, which may be attributed to distinct underlying co-morbidities and patient history (Table 1). A similar expression profile has been shown by others in diverse pathophysiological conditions, such as cancer [8,9,10,11,12,13], fibrosis [14,15,16], metabolic disorders [17,18,19], and arthritis [20,21,22,23,24,25,26].

### 3.2. TGFβ1 Stimulation Displays Differential WISP1 Expression Profile in Primary NHLF and DHLF

Transforming growth factor-β (TGFβ1) is the most potent, master regulator of fibrosis and inflammation [27,28]. However, due to its pleiotropic role in maintaining homeostasis, TGFβ1-blocking therapeutic modalities have caused severe adverse effects in clinical trials [29,30]. Given the challenges of targeting TGFβ1 for fibrogenesis, we aimed to assess the relationship between TGFβ1 and WISP1. Stimulation with TGFβ1 (1 ng/mL) yielded a time-dependent increase in the profibrotic gene signature with statistically significant effects noted at 24 h in both NHLF and DHLF (Appendix A). Furthermore, the effect of TGFβ1 was apparent on WISP1 gene expression as early as 2 h in DHLF (1.9-fold) and continued to increase till 24 h for both NHLF (8.3-fold) and DHLF (3.2-fold) (Appendix A).

After we established that the 24 h time point induced a fibroblast-to-myofibroblast transition (FMT) gene signature, we next wanted to determine if this is also true for multiple donor-derived primary fibroblast cell lines (*n* = 3/each) to gain confidence with respect to utilizing TGFβ1 as a positive control. The results from these experiments revealed that TGFβ1 stimulation significantly elevated profibrotic genes, such as collagen type 1 α1 chain (COL1A1), collagen type 3 α1 chain (COL3A1), α smooth muscle actin (ACTA2/αSMA), fibronectin (FN1), and inflammatory genes, such as interleukin-6 (IL6), in all 3 donor cell lines, with expected donor-to-donor variability for NHLF (Figure 2A), DHLF (Figure 2B), and NHDF (Figure 2C). However, to our surprise, chemokine ligand 2/monocyte chemoattractant protein-1 (CCL2/MCP1) was significantly downregulated in NHLF (Figure 2A) and DHLF (Figure 2B), but not in NHDF (Figure 2C), a striking difference between the lung versus the dermal fibroblast response to TGFβ1. Furthermore, the TGFβ1 time course studied in NHLF and DHLF confirmed these findings and revealed a biphasic response to CCL2 gene expression, which increased until 4 h, followed by a clear decline until 24 h (Appendix A). Overall, these results suggest that WISP1 operates downstream of TGFβ. Given that TGFβ levels are markedly elevated in the fibrotic disease state, as shown by others [31,32], this could partly explain the significant elevation of WISP1 observed in DHLF (Figure 1A). Interestingly, similar findings have been reported in hepatic stellate cell lines [16].

### 3.3. WISP1 Promotes Fibroblast Cell Proliferation

Fibroblast cell proliferation is a key mechanism in the induction and progression of fibrosis. Given our results showing that WISP1 is upregulated in IPF, next, we wanted to study the role of WISP1 in fibroblast proliferation. Retinoblastoma protein (Rb) is a tumor suppressor protein that regulates cell proliferation and the cell cycle. CDK-dependent phosphorylation of Rb (pRb) inactivates the protein and facilitates entry into the S-phase of the cell cycle, thereby promoting cell proliferation [33,34]. An immunofluorescence technique was utilized to detect and visualize pRb positive cells as a surrogate marker for cell proliferation (Figure 3A). Our results show that treatment with recombinant human WISP1 significantly increases cell proliferation in a concentration-dependent manner, with a maximum response peaking at approximately 30% pRb positive cells at the highest concentration compared to unstimulated control in primary human dermal fibroblasts (Figure 3B). Platelet-derived growth factor (PDGF) served as an internal positive control in all the experiments and yielded a robust fluorescent signal indicative of saturated cell-proliferation response at 100 nM with approximately 40% pRb positive cells (*p* < 0.0001 via one-way ANOVA) (Appendix A). Although, TGFβ1 treatment for 24 h did not increase cell proliferation at higher doses, at lower doses (0.0625 ng/mL), it significantly increased pRb positive cells (*p* < 0.0001 via one-way ANOVA) (Appendix A).

### 3.4. WISP1 Drives Initiation of Pro-Inflammatory Cytokines Induction

To study whether the WISP1 effect on fibrogenesis is direct or indirect, we first examined its role in fibrosis initiation by assessing the profibrotic gene signature, similar to TGFβ1 stimulation studies, as shown previously (Figure 2). TGFβ1 (1 ng/mL) served as a positive control as stimulation for 24 h robustly initiated fibrotic gene induction (Figure 2). To our surprise and contrary to what others have reported [35], we did not observe any direct effect of WISP1(1000 nM) treatment in initiating the induction of profibrotic genes increase, particularly for collagens such as COL1A1, COL3A1, and ACTA2, and FN1 in NHLF (Figure 4A), IPF-DHLF (Figure 4B), and NHDF (Figure 4C). In addition, WISP1 stimulation had no effect on collagen protein levels (Appendix A), further confirming our findings. However, interestingly, stimulation of NHLF and DHLF with WISP1 significantly elevated pro-inflammatory cytokines, particularly IL6 (1.5 and 2.1-fold, respectively) and CCL2/MCP1 (5 and 1.3-fold, respectively) as compared to control (Figure 4A,B). Among the three main fibroblast cell lines, WISP1 had the most significant impact on IL6 and CCL2 in NHDF, showing approximately 8.9-fold and 13-fold increases compared to the unstimulated control (Figure 4C). The levels of IL6 and CCL2 in the cell-culture supernatant increased in a concentration-dependent fashion when compared to the control (Figure 4D). In addition, these results also highlight that gene-expression analysis (Figure 4C) confirms and aligns well with the observed secreted protein levels (Figure 4D).

Notably, our findings reveal that, like TGFβ1, WISP1 stimulation also enhances its own expression in NHLF (6-fold) and DHLF (1.3-fold) (Figure 4A,B), indicating a possible positive auto-regulatory mechanism. This autocrine effect of WISP1 was not observed in NHDF (Figure 4C), highlighting the self-regulatory mechanistic disparity amongst different organ-derived fibroblast subpopulations.

### 3.5. WISP1 with TGFβ1 Synergistically Induced Pro-Inflammatory Cytokines

Although WISP1 did not directly initiate fibrosis, we aimed to investigate its potential role in disease progression, similar to findings observed in liver fibrosis in vivo [7]. We reasoned that establishing a simple in vitro system would be beneficial. We first treated cells with TGFβ1 to initiate profibrotic gene expression and mimic the pre-existing fibrotic environment, and then followed by WISP1 stimulation to further assess whether WISP1 works in concert with TGFβ1 to facilitate disease progression in human fibroblasts (Figure 5A). Our results show that gene expression of pro-inflammatory cytokines, particularly IL6 was synergistically upregulated when NHLF (Figure 5B) and NHDF (Figure 5D) were treated with TGFβ1 + WISP1 as compared to either TGFβ1 or WISP1 only. The data also reveal an additive effect of the TGFβ1 + WISP1 treatment condition on CCL2 gene expression in NHLF (Figure 5B) and NHDF (Figure 5D), while similar effects were not observed in DHLF (Figure 5C). Surprisingly, similar to our previous findings, TGFβ1 significantly reduced, while in contrast, WISP1 increased CCL2 gene expression (Figure 2B and Figure 4B). For IL6, although either TGFβ1 or WISP1 alone facilitated IL6 induction, no additive effects were observed when treated in combination in DHLF (Figure 5C).

We also wished to assess the compound effect of TGFβ1 + WISP1 on other structural and extracellular genes, given their direct role in disease pathophysiology. To our surprise, the combination of TGFβ1 + WISP1 also did not show any significant alterations in the profibrotic gene expression, such as COL1A1, COLl3A1, ACTA2, and FN1, in all three primary human fibroblast cell lines (Appendix A–C). Further, given the positive-self regulatory function of WISP1 on itself, in conjugation with the TGFβ1-mediated WISP1 expression seen earlier in our studies (Figure 4A,B), WISP1 expression was notably upregulated upon combinatorial treatment in both NHLF (Appendix A) and NHDF (Appendix A) but not in DHLF (Appendix A). To our knowledge, these findings are the first to demonstrate the cooperative interaction between TGFβ1 and WISP1 in vitro, inducing inflammatory responses and showing differences between healthy and diseased fibroblasts, as well as between lung and skin fibroblasts, indicating organ-specific fibroblast heterogeneity. Given the highly dynamic and multifarious nature of fibroblasts, we next utilized NanoString^®^ for the transcriptional profiling of human lung and dermal fibroblasts to better understand the relationship between WISP1 and fibrogenesis by identifying novel WISP1-mediated profibrotic and inflammatory targets.

### 3.6. WISP1-Mediated Novel-Fibrotic Gene Signature in Primary Human Dermal Fibroblasts

Although we did not observe any direct effect of WISP1 on profibrotic genes, we next employed NanoString nCounter^®^ technology to further understand the role of WISP1 in the initiation and progression of fibrogenesis by treating primary human lung and dermal fibroblasts with either WISP1 (initiation) or in combination with TGFβ1 (progression).

NanoString^®^ transcriptional profiling analysis revealed novel WISP1-induced gene targets. Amongst these, 117 genes were significantly upregulated while 34 genes were downregulated (|FC| > 1.5; adj. *p*-value < 0.1) upon WISP1 (1000 nM) stimulation for 24 h in NHDF (Appendix A). Of the 117 upregulated genes, our results demonstrated a highly interconnected group of inflammatory genes, such as interferon α (IFNα)-inducible protein, IF127, IFI6, and IFI35, ISG15, and XAF1 (Figure 6A). Furthermore, WISP1 also induced the expression of another cluster of interleukin-related inflammatory markers, such as IL12RB2, IL18R1, IL1R1, IL1RAP, IRAK3, IL6, and IL6ST. In addition, genes involved in chemokine signaling, such as CCL2, CXCL2, CXCL8, CXCL10, and CCR4, were amongst the most significantly associated genes, suggesting an inflammatory axis of WISP1 biology. Cellular adhesion molecules, like ICAM1 and VCAM1, that are crucial for leukocyte infiltration and inflammation were also notably upregulated. Similar to these findings, others have also shown WISP1-mediated upregulation of VCAM1 [10,36,37] and ICAM1 [10,38] are crucial for cancer cell migration, invasion, and wound healing. In addition, integrins have been identified as receptors for most WISP1 biological functions [5,39]. For the first time, our results here demonstrate that WISP1 can, in turn, regulate the expression of integrins, particularly α4 (ITGA4), α5 (ITGA5), and ILK (Appendix A). Overall, our NanoString^®^ analysis indicates that WISP1 significantly promotes inflammation by activating genes associated with inflammatory signaling pathways, thus contributing to the disease pathology.

Interestingly, similar to what others have shown, WISP1 drives matrix metalloprotease expression, particularly MMP1 and MMP3 expression, which could further modulate fibrosis [5]. Other potential genes involved in cell proliferation and cell cycle regulation, such as E2F2 and CCNA2, along with MAPK11, were markedly elevated by WISP1 in dermal fibroblasts. These results further confirm and strengthen our previous results (Figure 3A,B) by providing a molecular basis for WISP1-mediated cellular proliferation in NHDF. In addition to the genes mentioned above, a list of all WISP1-induced genes, with their respective fold change (FC) value, log2 FC, adj. *p*-value, and other statistical parameters, is provided in the Appendix A.

To further gain insight into the cooperative interaction of TGFβ1 + WISP1 in disease progression, we next sought to determine other synergistically regulated fibrotic-transcriptional signatures, elicited by these two mediators in NHDF. We identified 24 synergistically upregulated genes, while 2 genes were synergistically downregulated (|FC| > 1.5; adj. *p*-value < 0.1) in the TGFβ1 + WISP1 condition as compared to either TGFβ1 only or WISP1 only (Figure 6B,D and Appendix A). In addition to IL6 and CCL2, the gene set comprises transcripts associated with growth factors, the extracellular matrix (ECM), inflammation and other fibrotic targets, such as collagen (COL4A1, COL7A1), growth factors (FGF2, VEGFA, CCN2), matrix metalloproteases (MMP1, MMP3), interleukins (IL1RAP, IL11), along with TGFβ receptor 1 (TGFBR1), TNF receptor superfamily 17 (TNFRSF17), hypoxia-inducible factor 1-α (HIF1α), serpin family E member 1 (SERPINE1), and serpin family H member 1 (SERPINH1), amongst others listed in the Appendix A. While WISP1 alone did not directly increase structural proteins like collagen, their levels were significantly upregulated in the presence of TGFβ1. This suggests an interdependency between WISP1 and TGFβ1 to produce a coordinated fibrogenic effect. Cyclin-dependent kinase inhibitor 2C (CDKN2C) was markedly downregulated in the TGFβ1 + WISP1 condition, suggesting another parallel yet indirect mechanism promoting WISP1-mediated cell proliferation. Furthermore, we also highlighted a unique and overlapping set of genes (Figure 6E), along with their associated pathways induced by different treatment paradigms (Figure 6C). TGFβ mediated fibrotic gene signature is illustrated in the Appendix A.

### 3.7. WISP1-Mediated Fibrotic Gene Signature in Primary Human Lung Fibroblasts

NanoString^®^ analysis revealed WISP1 induced novel gene targets. Among these genes, 331 were significantly upregulated while 13 were downregulated (|FC| > 1.5; adj. *p*-value < 0.1) in NHLF (Figure 7A and Appendix A). Amongst these, interleukin-related inflammatory markers, such as IL6, IL4, IL21R, IL6ST, IL1RAP, and IRAK3, were most notably upregulated. Similar to dermal fibroblasts, chemokine-signaling-associated genes, such as CCL2, CXCL2, CXCL8, and CXCL10, were also upregulated in lung fibroblasts as well, with the addition of CCL13 and CXCL11 in NHLF. Furthermore, our results also demonstrated a remarkably high degree of similarity in normal lung and dermal fibroblasts with respect to IFNα-inducible protein-related genes upon WISP1 stimulation, such as IFI6, IFI27, IFI35, ISG15, and ISG20 (Figure 7A and Appendix A). We observed a significant association of WISP1 with proliferative genes, such as MAPK1, MAPK9, MAPK11, MAP2K1, MK167, CDK4, and CCNA2, along with transcriptional factors, like E2F4 and S100A4, suggesting a direct role of WISP1 in promoting fibroblast cell proliferation, an important part of fibrosis (Figure 3). Surprisingly, and notably, a significant fold-change increase in 9 different collagen associated genes, such as COL1A1, COL4A1, COL5A1, COL5A3, COL6A3, COL6A5, COL7A1, COL10A1, and COL16A1, and FN1 was observed in contrast to our previous qPCR data (Appendix A), suggesting higher sensitivity of the NanoString^®^-based multiplex technique over conventional PCR and a potential direct action of WISP1 on fibroblasts in promoting fibrosis. Other profibrotic-related genes directly modulated by WISP1 included growth factors (VEGFA, EGFR, NEGF, FGF2, FLT1/VEGFR1, CCN2), cellular hypoxia inducible factors (HIF1A, HIF3A), cellular adhesion molecules (VCAM1, ICAM1), and serine protease inhibitor family members (SERPINE1, SERPINH1) (Appendix A). A recent study showed that WISP1 phosphorylates and activates EGFR to activate MIF, which, in turn, drives lung inflammation and remodeling [40].

Our previous data revealed that WISP1 is downstream of TGFβ1 (Figure 2A); however, another interesting observation from the transcriptomics data is that WISP1 can also modulate TGFβ signaling by increasing TGFβ-receptor TGFBR1, TGFBR2, and TGFB1I1 expression in primary lung fibroblasts (Appendix A). However, similar results were not observed with conventional qPCR gene expression analysis, again highlighting the tool-specific disparity with respect to assay sensitivity and specificity. WISP1 can further self-regulate itself via augmenting its integrin receptor expression (ITGB1, ITGB5, ITGA4, ILK) or by promoting MMP expression (MMP1, MMP7, MMP10, MMP13, MMP16) in NHLF to drive cell migration via distinct mechanisms, which can further enhance fibrogenesis [5]. A detailed list of all the genes is provided in the Appendix A, with their respective FC value, log2 FC, adj. *p*-value, and other statistical parameters.

To further examine the synergistically regulated fibrotic-gene profiling, we identified a total of 47 differentially regulated genes, out of which 7 were synergistically upregulated, while 27 were synergistically downregulated (|FC| > 1.5; adj. *p*-value < 0.1) in the TGFβ1 + WISP1 condition in comparison to both TGFβ1 only and WISP1 only (Figure 7B,D and Appendix A). We found 4 out of the 7 upregulated genes stimulated in NHLF were exactly similar to the NHDF gene signature, including transcripts associated with growth factor (CCN2/CTGF), collagen (COL7A1), and interleukins (IL1RAP, IL6), highlighting a high degree of congruence with respect to collective TGFβ1 + WISP1-induced signaling in non-diseased lung and dermal primary human fibroblasts. Other synergistically upregulated genes were ISG15, VCAM1, and XAF1. As a detailed interpretation of all the genes is beyond the scope of this paper, a list of all the significantly modulated genes along with the fold change and *p*-values is provided in the Appendix A. Out of the unique genes, tight-junction protein 2 (TJP2), catalase (CAT), death-associated protein kinase (DAPK1), and CDKN2C were amongst the unique set of genes that were significantly downregulated (Appendix A). Chronic inflammatory lung diseases are characterized by oxidative stress due to dysregulation in the oxidant and antioxidant system that can not only drive disease progression but can also determine therapeutic outcomes [41,42]. Catalase acts as an endogenous antioxidant enzyme that helps catalyze the breakdown of hydrogen peroxide into water and oxygen. Decreased catalase expression at both mRNA and protein levels has been reported in humans and in a mouse bleomycin fibrosis model [43], and our results highlight the combined role of WISP1 and TGFβ1 in declined catalase activity during fibrogenesis. In addition, the TJP2 gene encodes for a tight junction-associated protein, namely, zonula occludens-2 (ZO-2) and dysregulation in the TJ protein leads to barrier dysfunction with compromised integrity, a common feature of IPF [44]. Furthermore, we also highlighted a unique and overlapping set of genes (Figure 7E) along with their associated pathways induced by different treatment paradigms (Figure 7C).

### 3.8. WISP1-Mediated Fibrotic Gene Signature in Primary Human Diseased (IPF) Lung Fibroblasts

To better understand the role of WISP1 in fibrogenesis and to uncover the associated pathways, we also sought to identify gene transcripts in diseased (IPF) primary lung fibroblasts. Our NanoString^®^-based transcriptomics analysis revealed a novel molecular signature, amongst which 64 genes were significantly upregulated while 7 genes were downregulated (Figure 8A and Appendix A). These genes were filtered and trimmed down based on our stringent inclusion criteria (|FC| > 1.5; adj. *p*-value < 0.1) to only represent highly relevant effects in diseased human lung fibroblasts. Amongst these, IFNα-inducible protein-related genes, including IFI6, IFI27, IFI35, and ISG15 were also modulated in diseased human lung fibroblasts, analogous to both lung and dermal fibroblasts. In addition, our results also highlight a high degree of similarity in the WISP1-mediated gene signature; for instance, the chemokine-signaling-related genes expression signature (CCL2, CCL13, CXCL2, CXCL8, CXCL10, CXCL11) in DHLF (Appendix A) is exactly similar to what was observed in NHLF (Appendix A). Other highly correlated genes were integrins (ITGA4, ITGA5), cellular adhesion molecules (VCAM1, ICAM1), serine protease inhibitor family members (SERPINE1, SERPINH1), and hypoxia-inducible factors (HIF1A). However, fewer interleukin (IL6, IL1RAP), MMPs (MMP1), and growth factor-related (FLT4/VEGFR3) genes were observed in the diseased state (Appendix A) as compared to non-diseased lung fibroblasts (Appendix A), which could potentially be due to increased WISP1 leading to desensitization of the Wnt-1 associated downstream signaling pathway in an IPF-diseased state. Furthermore, consistent with our findings in lung fibroblasts, the expression of profibrotic genes, particularly collagen-associated genes, such as COL1A1, COL3A1, COL4A1, COL5A1, and COL5A3, was strongly associated with WISP1 in IPF-diseased fibroblasts but not in dermal fibroblasts, suggesting tissue-specific differences and heterogeneity in primary human fibroblasts (Appendix A).

In addition to the fibroblast-lineage-associated disparity in molecular signature, we also found a common cluster of a 31 highly significant core gene set, which were highly correlated in both diseased and non-diseased lung and dermal fibroblasts (Appendix A). These set of genes comprises of transcript associated with interferon-induced genes, such as 2′-5′-oligoadenylate synthase1 (OAS1) and X-linked inhibitor of apoptosis (XIAP)-associated factor 1 (XAF1) or NF-κB pathway-associated members, including transcription factor RelB (RELB) and tribbles pseudokinase 3 (TRIB3). Transcription factor RELB is a member of the NF-κB family, which has been utilized not only as a promising diagnostic and monitoring biomarker in renal fibrosis but has also been attributed to promote inflammatory cytokines expression to drive disease progression [45,46]. Accumulating evidence from the literature also points towards the crucial role of stress-response protein TRIB3 in pulmonary and renal fibrogenesis by promoting ECM deposition [47,48,49,50,51]. In contrast, another study reports that TRIB3 expression is markedly downregulated in IPF patients both at the transcript and protein level, and TRIB3 overexpression inhibits fibroblast activation and ECM deposition [52]. In addition, signaling pathways of a broad range of proinflammatory and profibrotic cytokines (such as IL6), interferon, as well as growth factors (such as VEGF, FGF), by canonical or non-canonical pathways, culminate downstream in the JAK/STAT pathway [53,54], and given that our data support direct modulation of these upstream modulators by WISP1, downstream transcription factor, such as STAT1, was also found to be significantly upregulated, further validating our findings. Furthermore, the role of the JAK/STAT pathway in fibrotic diseases has been extensively documented in the literature and small-molecule JAK/STAT inhibitors have been found to alleviate excessive inflammation and provide anti-fibrotic effects both in vitro and in vivo [54]. Here, our results reveal that WISP1 activates STAT1, which, at least in part, could contribute to heightened inflammation, influencing fibrosis pathophysiology. From the identified gene cluster, som unconventional fatty acid metabolism-related genes, such as fatty acid elongase 6 (ELOV6), fatty acid binding protein 5 (FABP5), and stearoyl-CoA desaturase (SCD), were also significantly associated with WISP1 in all the three lung and dermal primary fibroblast cell lines (Appendix A). Alteration in fatty acid composition and dysregulation in fatty acid metabolism has been identified in both a preclinical bleomycin-model and in IPF lung tissue, and plays a critical role in macrophage polarization, epithelial-to-myofibroblast transition (EMT), and fibroblast activation, suggesting a key role in disease pathophysiology [55]. The FABP5 gene encodes for fatty acid binding protein that is responsible for lipid trafficking metabolism and has been shown to aggravate pulmonary fibrosis through the Wnt-β-catenin pathway [56]. In contrast, and to our surprise, SCD1 and ELOVL6 expression was downregulated in IPF-lung tissues [57,58]. Although decreased ELOVL6 expression was specifically localized in alveolar type II (AT2) epithelial cells, our findings primarily focus on fibroblasts, indicating cell-type-specific differences. Furthermore, SCD mRNA expression was significantly upregulated in hepatic stellate cells (HSCs) that activate the Wnt-pathway by stabilizing the Frizzled/low-density lipoprotein (LDL) receptor-related protein (LRP) 5/6, promoting liver fibrosis [59]. In addition, although the role of fatty acid metabolism in IPF remains controversial, unsaturated fatty acids were found to be significantly upregulated in a bleomycin animal model as compared to control. This aligns with our results, as increased SCD-catalyzed fatty acid desaturation leads to increased unsaturated lipids [60]. Overall, our findings highlight the previously underappreciated role of dysregulated lipid metabolism and reveal its link with WISP1 in the pathogenesis of pulmonary fibrosis.

To gain better insight into the role of WISP1 in progression, we identified a total of 30 differentially modulated genes out of which 7 genes were synergistically upregulated, while 5 genes were synergistically downregulated (|FC| > 1.5; adj. *p*-value < 0.1) in the TGFβ1 + WISP1 condition upon comparison to both TGFβ1 only and WISP1 only in IPF-diseased fibroblasts (Figure 8B,D and Appendix A). Among these genes IL11, IL1RAP, ISG15, HLAF, CCR4, CCNA2, and COL7A1 were amongst the most significantly associated genes by conjoint TGFβ1 + WISP1 in IPF. In addition, DAPK1, endothelial PAS domain protein 1 (EPAS1), protein kinase AMP-activated non-catalytic subunit gamma 2 (PRKAG2), protein patched 1 (PTCH1), and RAS oncogene (RAB7B) were significantly downregulated (Appendix A). Patched-1 protein plays a critical role in fibrogenesis and has been shown to be significantly downregulated both in vitro in hepatic stellate cells and in vivo in a CCl4-induced rat fibrogenesis model, due to PTCH1 hypermethylation [61]. Furthermore, DAPK1 is an interferon-induced serine/threonine kinase involved in programmed cell death and was significantly decreased by TGFβ1 + WISP1 treatment, making fibroblasts resistant to apoptosis, which further contributes to disease pathophysiology and fibrogenesis in both non-diseased and IPF primary lung fibroblasts. Furthermore, we also highlighted a unique and overlapping set of genes (Figure 8E), along with their associated pathways induced by different treatment paradigms (Figure 8C).

## 4. Discussion

CCN family members have attracted considerable attention due to their pleiotropic role in human health and diseases. Despite the extensive research and increased attention, WISP1 still remains one of the understudied members of the CCN family. Thus far, the majority of research efforts have been directed towards CCN2/CTGF, but the discontinuation of the phase-3 ZEPHYRUS-1 study of Pamrevlumab for IPF has shifted the spotlight towards other promising CCN members as therapeutic targets. Over the past two decades, many studies have suggested that WISP1 plays a profibrotic role by driving disease progression rather than being involved in the initiation of fibrosis. Accordingly, as therapeutic strategies targeting WISP1 are currently at Phase 1 clinical trials, our results, for the first time, uncover the molecular signature of WISP1 profibrotic biology in a cell and disease-specific context, i.e., dermal vs. lung fibroblasts and between healthy vs. IPF-diseased states. Importantly, our current study indicates that WISP1 is upregulated at both transcript and protein levels in IPF-diseased primary lung fibroblasts compared to non-diseased primary lung. TGFβ is the master regulator of fibrosis, and our results also highlight a feedback-loop between TGFβ-WISP1, where the two proteins positively regulate each other.

Furthermore, fibroblast proliferation is one of the key mechanisms involved in fibrogenesis and our study demonstrates WISP1 markedly induces human fibroblast cell proliferation, setting the stage for TGFβ (Figure 9). In addition, our transcriptomics data reveal a mitogenic and cell-cycle-associated molecular signature, including MAPKs, CDK4, CCNA2 and marker of proliferation Ki-67 (MK167), along with transcriptional factor, like E2F4, further corroborating that WISP1 induces fibroblast proliferation. Results from the current study also indicate that the inflammatory gene signature exhibits high sensitivity to WISP1 as it positively regulates chemokine signaling, interleukin-related inflammatory markers, and interferon-stimulated genes in human fibroblasts.

Given the biological complexity and aggressive nature of IPF, it is not surprising that narrowly targeting one molecule fails to meet the clinical endpoint. A multimodal therapeutic strategy targeting multiple molecular pathways may yield better patient outcomes. Our study reveals the molecular underpinnings of the multifaceted role of WISP1 in driving various cellular processes, ranging from cell proliferation, migration, to inflammation. Therefore, targeting WISP1 could yield positive outcomes. Research and experiments are underway in our laboratory to delve deeper into the downstream mechanisms of action and extend this work in vivo, further testing and demonstrating translatability.

While the current study aims at unraveling WISP1 biology in an IPF setting focusing on human fibroblasts, our study has certain limitations. Firstly, it is critical to acknowledge that fibrosis is a complex disease with spatial and temporal heterogeneity due to the highly dynamic cellular state. Here, we are studying primary fibroblasts in isolation, which fails to capture the paracrine effects of WISP1 and crosstalk between fibroblasts and tissue-resident cells, including but not limited to epithelial cells, endothelial cells, as well as other immune cells, such as macrophages and neutrophils. We are currently establishing co-culture systems that would shed more light on both the autocrine and paracrine effects of WISP1. Second, as these results represent only the initial investigation of the WISP1-induced molecular signature in primary dermal and lung fibroblasts, future work aims to increase the translatability of these findings by utilizing preclinical animal models to better recapitulate disease biology. Furthermore, future work in our lab will also focus on uncovering gene-maps based on the cell type in native animal tissues by utilizing spatial transcriptomics. Nonetheless, to our knowledge these are the first results that delineate the molecular underpinnings of WISP1-mediated fibrosis initiation and progression in primary human fibroblasts. In addition, our transcriptomics results must also be validated at the protein level to better correlate these findings. However, the IL6 and CCL2 messages at the transcript levels positively correlate with secreted protein levels, suggesting no disconnect between mRNA and protein levels. While here we only focused on lung and dermal fibroblasts, to better understand whether WISP1 has a direct or indirect role in promoting fibrosis, more studies are underway to assess its role in other organ-derived fibroblasts, particularly hepatic stellate cells, cardiac fibroblasts, and renal fibroblasts. Furthermore, biomanufacturing of active macromolecules, such as recombinant proteins and antibodies, has significantly advanced over time and is a key process in the pharmaceutical industry. However, some proteins remain difficult to express in conventional CHO or HEK cell-based systems due to several factors, including improper folding, degradation, oligomerization from overproduction, and post-translational modification. These issues can result in low yields or sometimes inactive products. WISP1 is one such ‘difficult-to-express’ protein in cell-based systems, often resulting in inconsistent yield and activity. To address these challenges, we are utilizing a lentivirus-mediated approach to mimic the endogenously upregulated WISP1 expression seen in diseased states. This method not only facilitates WISP1 production but also relies on cellular machinery to apply the necessary post-translational modifications for proper cellular functions, thereby bypassing the need for recombinant proteins. Furthermore, this lentivirus-mediated approach will provide us with the flexibility to explore the domain-specific functional effects of WISP1, as producing domain-specific recombinant proteins can be challenging. Overall, the current study underscores the identification of a novel gene-map to gain better insights into WISP1 profibrotic function and provides a foundation for future investigation, enabling the development of therapeutic strategies for fibrosis. It is critical to emphasize that further research is essential to delve deeper into the mechanism of action, particularly in understanding the diverse functional roles of WISP1 in organ-specific inflammation and fibrogenesis.

## Figures and Tables

**Figure 1 cells-13-02005-f001:**
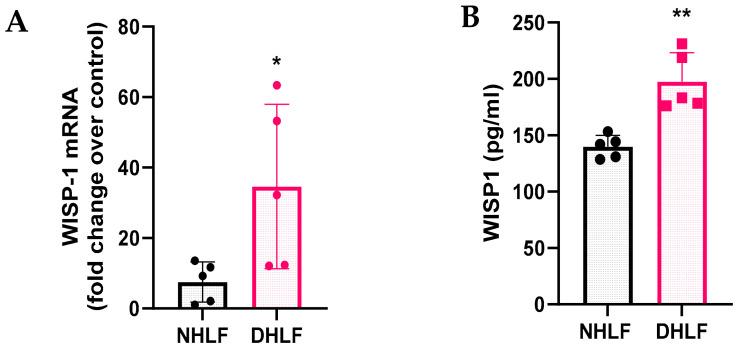
WISP1 is endogenously upregulated in primary human DHLF. WISP1 transcript and secreted protein are expressed in NHLF and DHLF. DHLF exhibit significantly higher (**A**) WISP1 mRNA transcript and (**B**) secreted WISP1 protein in the condition media, detected by RT-qPCR and ELISA, respectively. Each data point represents one donor-derived primary cell line for both NHLF (*n* = 5 donors) and DHLF (*n* = 5 donors), with each experiment performed in triplicates and represented as mean ± SD. Statistical analysis was performed using paired *t*-test and significance denoted as * *p* < 0.5 or ** *p* < 0.01 versus healthy NHLF, as shown.

**Figure 2 cells-13-02005-f002:**
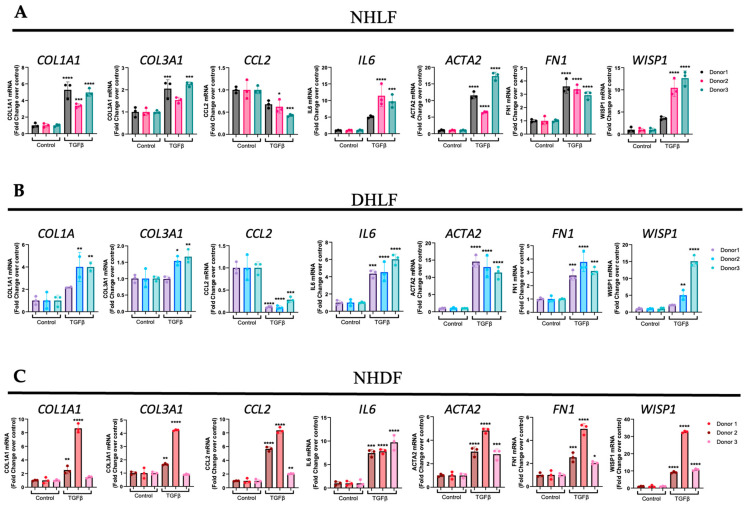
TGFβ induces WISP1 expression in primary lung and dermal fibroblasts. Stimulation with TGFβ (1 ng/mL) for 24 h significantly increases COL1A1, COL3A1, IL6, ACTA2, FN1, and WISP1 mRNA as compared to control in (**A**) NHLF, (**B**) DHLF, and (**C**) NHDF for 3 respective donors (*n* = 3/each) analyzed by RT-qPCR. Notably, TGFβ decreased CCL2 in NHLF and DHLF, except in NHDF. Each experiment was independently performed in triplicates with three biological replicates (*n* = 3) for each donor and represented as mean ± SD. Statistical analysis was performed using one-way ANOVA with Tukey’s post hoc analysis vs. vehicle control for respective donors. * denotes *p* < 0.5, ** denotes *p* < 0.01, *** denotes *p* < 0.001, **** denotes *p* < 0.0001 versus the unstimulated control, as shown.

**Figure 3 cells-13-02005-f003:**
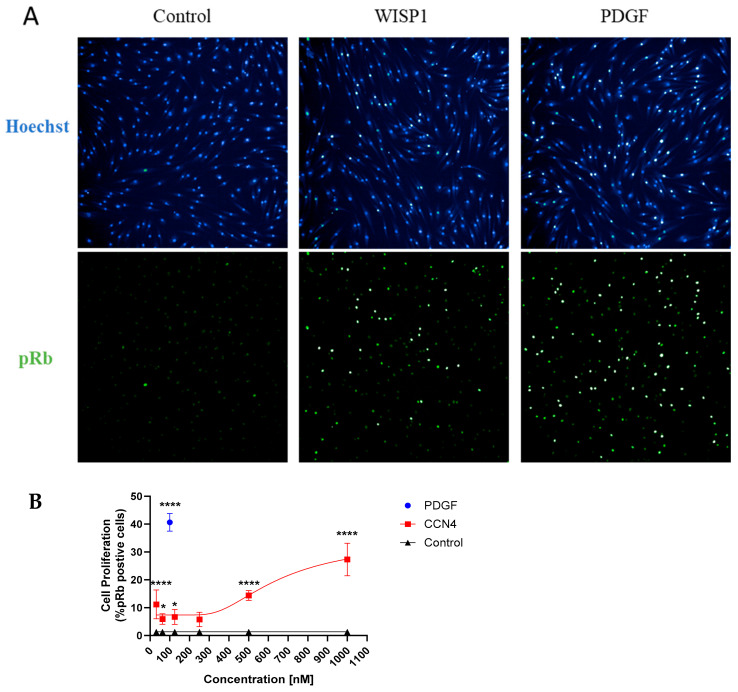
WISP1 promotes a striking increase in primary human dermal fibroblast cell proliferation. (**A**) Treatment with WISP1 (1000 nM) shows a striking increase in cell proliferation (*p* < 0.0001 via one-way ANOVA with Tukey’s post hoc analysis) detected by increased pRb positive cells as compared to control in NHDF. Representative 10× images were captured with the top image representing Hoechst nuclear stain, while the bottom panel represents pRb green fluorescence. PDGF-BB (100 nM) was used as an internal positive control to ensure assay reliability. The PDGF-BB dose response curve is shown in the Appendix A. (**B**) % pRb positive cells were calculated by dividing pRb positive cells over total number of cells calculated by nuclear Hoechst staining. Three independent experiments (*n* = 3) were performed in triplicates and represented as mean ± SD. Statistical analysis was performed using one-way ANOVA with Tukey’s post hoc analysis versus the vehicle-control condition and significance denoted as * *p* < 0.5, **** *p* < 0.0001.

**Figure 4 cells-13-02005-f004:**
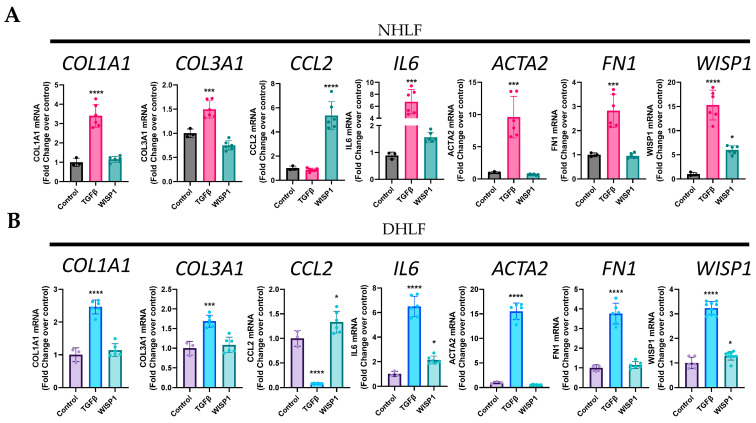
WISP1 increases CCL2 and IL6 in primary fibroblasts. Stimulation with WISP1 (1000 nM) for 24 h significantly increases CCL2 and IL6 gene expression in (**A**) NHLF, (**B**) DHLF, and (**C**) NHDF as compared to control. TGFβ (1 ng/mL) for 24 h served as a positive control and significantly increases COL1A1, COL3A1, IL6, ACTA2, FN1, and WISP1 versus vehicle control. (**D**) Stimulation with increasing WISP1 concentrations for 24 h initiates concentration-dependent increase in CCL2 and IL6 protein levels, detected via ELISA in the cell-supernatant of NHDF. Each experiment performed in duplicates with three biological replicates (*n* = 3) and represented as mean ± SD. Statistical analysis was performed using one-way ANOVA with Tukey’s post hoc analysis vs. vehicle control for respective donors. * denotes *p* < 0.5, ** denotes *p* < 0.01, *** denotes *p* < 0.001, **** denotes *p* < 0.0001 versus the vehicle control, as shown.

**Figure 5 cells-13-02005-f005:**
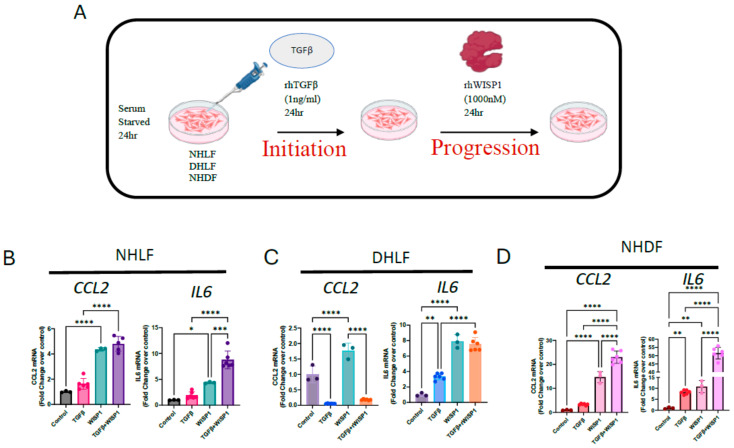
Recombinant WISP1 synergistically induces inflammatory markers along with TGFβ. (**A**) Primary human fibroblasts were stimulated with TGFβ (1 ng/mL) for 24 h for initiation, followed by WISP1 (1000 nM) for another 24 h as per the experimental layout. CCL2 and IL6 mRNA expression was significantly increased in (**B**) non-diseased lung and (**D**) dermal fibroblasts, but not in (**C**) IPF-diseased lung fibroblasts. Statistical analysis was performed using one-way ANOVA with Tukey’s post hoc analysis vs. vehicle control for respective donors. * denotes *p* < 0.5, ** denotes *p* < 0.01, *** denotes *p* < 0.001, **** denotes *p* < 0.0001 versus the unstimulated control, for three biological replicates (*n* = 3) and represented as mean ± SD.

**Figure 6 cells-13-02005-f006:**
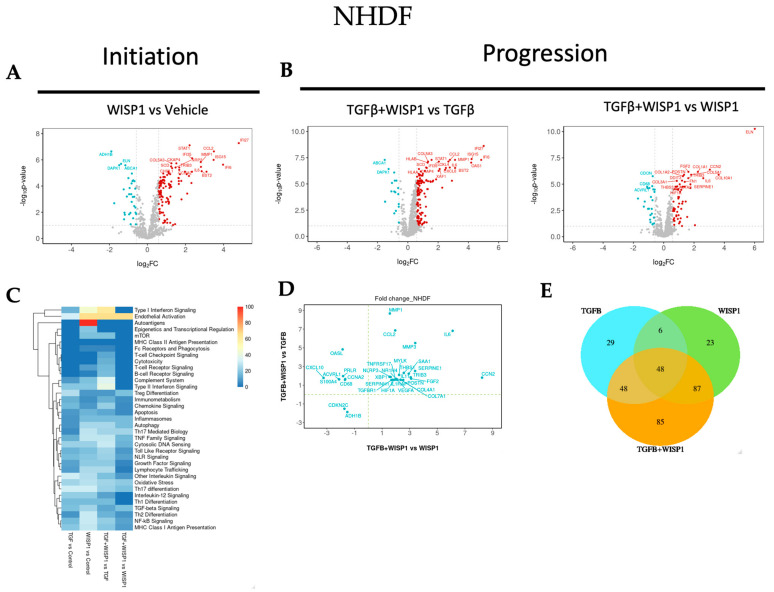
WISP1 fibrotic gene signature in dermal fibroblasts. The volcano plot illustrates fibroblast genes upregulated (red) and downregulated (blue) upon stimulation with either (**A**) WISP1 alone or (**B**) in conjugation with TGFβ as compared to vehicle control (PBS) in NHDF (*n* = 3). The plots representing statistically significant gene with |FC| > 1.5 and adj. *p*-value < 0.1 are plotted with the x-axis representing log2 fold change (FC) versus the y-axis −log10 *p*-value. (**C**) The heatmap represents pathways significantly associated with the corresponding treatment condition as percent of genes influenced and the list of genes used in the pathway analysis is highlighted in Appendix A. (**D**) Scatterplot showing log2 fold change for the differentially expressed genes when comparing WISP1 + TGFβ treatment with WISP1 (x-axis) versus TGFβ (y-axis). (**E**) Venn diagram representing unique and overlapping set of genes between the WISP1, TGFβ, and WISP1 + TGFβ treatment paradigms. A comprehensive lists of genes is provided in the Appendix A.

**Figure 7 cells-13-02005-f007:**
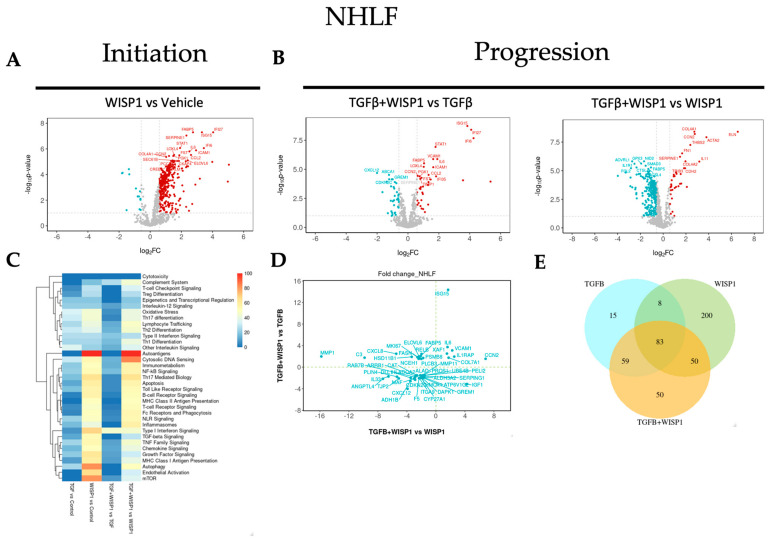
WISP1 fibrotic gene signature in lung fibroblasts. The volcano plot illustrates fibroblast genes upregulated (red) and downregulated (blue) upon stimulation with either (**A**) WISP1 alone or (**B**) in conjugation with TGFβ as compared to vehicle control (PBS) in NHLF (*n* = 3). The plots representing statistically significant gene with |FC| > 1.5 and adj *p*-value < 0.1 are plotted with the x-axis representing log2 fold change (FC) versus the y-axis −log10 *p*-value. (**C**) The heatmap represents pathways significantly associated with the corresponding treatment condition as percent of genes influenced and the list of genes used in the pathway analysis is highlighted in Appendix A. (**D**) Scatterplot showing log2 fold change for the differentially expressed genes when comparing WISP1 + TGFβ treatment with WISP1 (x-axis) versus TGFβ (y-axis). (**E**) Venn diagram representing unique and overlapping set of genes between the WISP1, TGFβ, and WISP1 + TGFβ treatment paradigms. A comprehensive lists of genes is provided in the Appendix A.

**Figure 8 cells-13-02005-f008:**
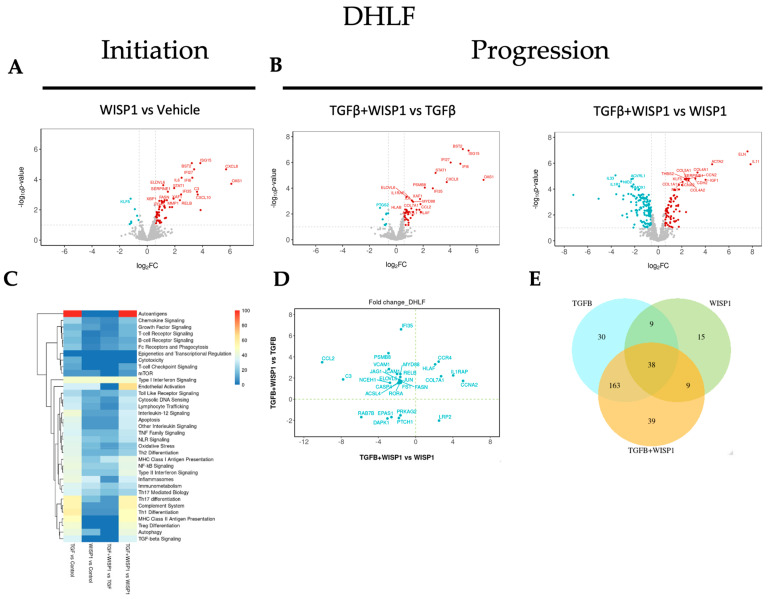
WISP1 fibrotic gene signature in IPF-diseased fibroblast. The volcano plot illustrates fibroblast genes upregulated (red) and downregulated (blue) upon stimulation with either (**A**) WISP1 alone or (**B**) in conjugation with TGFβ as compared to vehicle control (PBS) in DHLF (*n* = 3). The plots representing statistically significant gene with |FC| > 1.5 and adj *p*-value < 0.1 are plotted with the x-axis representing log2 fold change (FC) versus the y-axis −log10 *p*-value. (**C**) The heatmap represents pathways significantly associated with the corresponding treatment condition as percent of genes influenced and the list of genes used in the pathway analysis is highlighted in Appendix A. (**D**) Scatterplot showing log2 fold change for the differentially expressed genes when comparing WISP1 + TGFβ treatment with WISP1 (x-axis) versus TGFβ (y-axis). (**E**) Venn diagram represents unique and overlapping set of genes between the WISP1, TGFβ, and WISP1 + TGFβ treatment paradigms. A comprehensive list of genes is provided in the Appendix A.

**Figure 9 cells-13-02005-f009:**
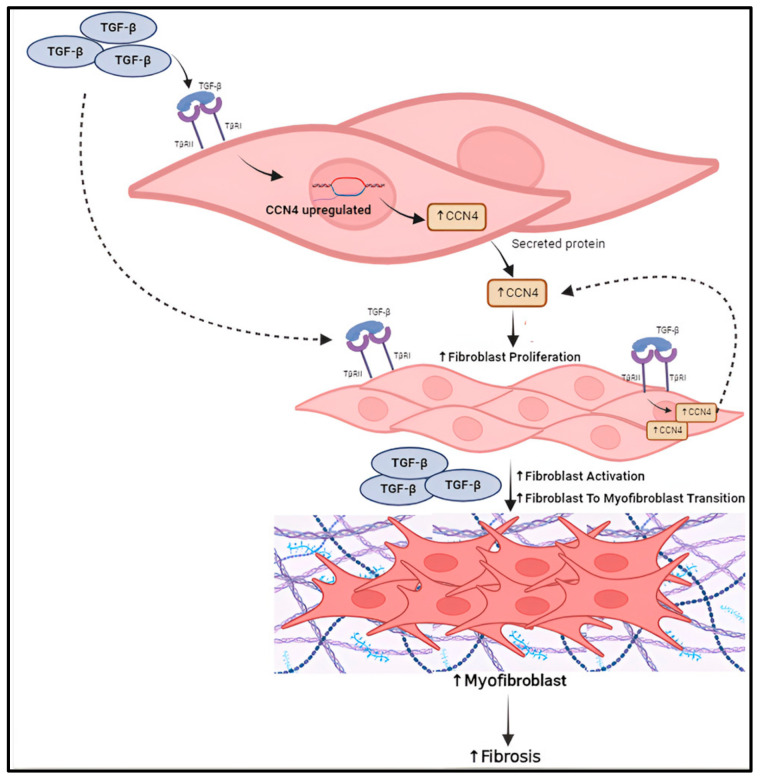
Schematic summary of WISP1-TGFβ crosstalk in regulation of fibrosis progression.

**Table 1 cells-13-02005-t001:** Clinical characteristics of donors.

Characteristics: Mean ± SD (Range)	NHLF (N = 5)	DHLF (N = 5)	NHDF (N = 3)
Age (years)	57.3 ± 14.3 (45–73)	65.6 ± 15.8 (52–83)	38.3 ± 9.1
Height (Inches)	61.3 ± 7.1 (55–70)	68 ± 0 (68–68)	62.3 ± 3.21 (60–66)
Weight (Kg)	80.1 ± 17.3 (60.7–108.4)	81.2 ± 6.6 (74.8–88)	67.8 ± 5.9 (61.2–69.9)
Sex (M/F)	2/3	5/0	0/3
Race	C (5)	C (5)	C (1), B (1), H (1)
Diabetes	0	2	0
Heart Disease	0	0	0
Hypertension	2	1	0
Alcohol	1	2	0
Smoke	0	1	0

Primary human non-diseased or diseased (IPF) lung and dermal cell lines derived from the enlisted patients. NHLF: Normal Human Lung Fibroblast, DHLF: Diseased Human Lung Fibroblast, NHDF: Normal Human Dermal Fibroblast. F: Female, M: Male, C: Caucasian, B: Black, H: Hispanic.

## Data Availability

All the relevant data are within the article and Appendix A. Questions and queries can be directed to the corresponding author.

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
