# Peer review of "Stage-Dependent Fibrotic Gene Profiling of WISP1-Mediated Fibrogenesis in Human Fibroblasts"

_cells, 2024, doi:10.3390/cells13232005_

Round 1

Reviewer 1 Report

Comments and Suggestions for Authors

The authors examine WISP1 (CCN4), a matricellular protein implicated in fibrosis with mechanisms that remain largely unclear. They identify a novel WISP1-associated profibrotic gene signature in primary human dermal and lung fibroblasts from both healthy individuals and IPF patients. The authors discuss WISP1’s upregulation in IPF fibroblasts, its downstream relationship with TGF-β, and its role in promoting fibroblast proliferation, as well as in inducing IL6 and CCL2 expression. A combined TGF-β and WISP1 in vitro model further demonstrates WISP1’s role in fibrotic progression. These findings reveal distinct molecular profiles of WISP1 in fibrosis, contributing to IPF-specific disease pathways. I believe this work aligns well with the scope of *Cells*, though revisions are necessary to address the following concerns.

Major Concerns/General comments

1. The authors propose that WISP1 alone does not upregulate pro-fibrotic genes in primary human lung fibroblasts but may function in synergy with TGF-beta, suggesting that WISP1 could be downstream of TGF-beta signaling. To substantiate this claim, it is essential for the authors to evaluate the effect of TGF-beta in both control and WISP1 knockout or knockdown cells. Relying solely on the combined addition of TGF-beta and WISP1 to enhance pro-fibrotic gene expression does not convincingly demonstrate that WISP1 functions downstream of TGF-beta signaling.

2. Previous studies suggest that WISP1’s primary source may not be fibroblasts but rather ATII cells in the lung (https://pmc.ncbi.nlm.nih.gov/articles/PMC2662540 ). In Figure 1A, WISP1 protein levels show only a slight increase in DHLF, with a relatively low concentration (~0.2 ng/mL). The authors should discuss whether fibroblast-derived WISP1 has a significant role in fibrosis compared to epithelial cell-derived WISP1.

3. The authors do not observe a direct effect of WISP1 (1000 nM) in inducing pro-fibrotic gene expression at the RNA level. Two alternative explanations should be considered: (1) WISP1’s effect on RNA might be rapid, making the 24-hour observation point too late. A 12-hour time point should be included; (2) in some cases, collagen protein levels may be more responsive than RNA levels, especially in HLF. The authors should directly measure ECM protein levels to verify that WISP1 alone does not induce fibrosis. Performing a hydroxyproline assay is recommended to assess collagen deposition directly.

Minor Concerns/Specific comments

1. In Figure 1A, qPCR data: If NHLF is used as the control group for qPCR fold change, its average should be 1, not ~5 as currently shown. Please double-check the data processing, as WISP1 is generally expected to increase 4-10 fold compared to control, rather than 20-60 fold as shown in Panel A. Clearly indicate the type of error bars (S.D. or S.E.M.) in the figure legend (applies to all figures).

2. In Figure 4, panel D lacks a label.

3. Figures 6, 7, and 8C (pathways analysis): Clarify the meaning of the heat gradient—is it based on p-adjusted values for each pathway or the percentage of genes influenced by each condition? Additionally, I recommend listing the specific genes used in pathway/enrichment analyses for each group in Panel C (e.g., TGFB vs. Control). For Panel D, indicate which genes are highlighted as ‘TGFB+WISP1 vs TGFB’ and consider marking these genes in Panel C. Supplementary tables should clearly correspond to Panel C’s gene lists for ease of reference.

4. For data availability, it would be beneficial if the authors could upload their raw data to GEO or, at minimum, provide the counts file as supplementary material.

Author Response

Reviewer #1

Major Concerns/General comments

  1. The authors propose that WISP1 alone does not upregulate pro-fibrotic genes in primary human lung fibroblasts but may function in synergy with TGF-beta, suggesting that WISP1 could be downstream of TGF-beta signaling. To substantiate this claim, it is essential for the authors to evaluate the effect of TGF-beta in both control and WISP1 knockout or knockdown cells. Relying solely on the combined addition of TGF-beta and WISP1 to enhance pro-fibrotic gene expression does not convincingly demonstrate that WISP1 functions downstream of TGF-beta signaling.

Response 1- We appreciate the comment and thank the reviewer for reviewing our manuscript. WISP1 knockout or knockdown would be an excellent strategy as suggested. However, similar to our findings, numerous literature evidence also indicates that WISP1 is downstream of TGFβ [1]. For instance, Klee and colleagues demonstrated that WISP1 is downstream of TGFβ and TNF-α in primary human lung fibroblast [2]. Furthermore, these results were also confirmed in hepatic stellate cells [3,4], kidney tubular epithelial cells [5] and chondrocytes [6], where authors demonstrate that TGF-β drives WISP1 expression at both transcript and protein levels. 

These findings were also validated via WISP1 KD approach, as suggested by the reviewer  [4,5]. Overall, there is substantial literature evidence supporting our findings that WISP1 is downstream of TGFβ. 

  1. Previous studies suggest that WISP1’s primary source may not be fibroblasts but rather ATII cells in the lung (https://pmc.ncbi.nlm.nih.gov/articles/PMC2662540 ). In Figure 1A, WISP1 protein levels show only a slight increase in DHLF, with a relatively low concentration (~0.2 ng/mL). The authors should discuss whether fibroblast-derived WISP1 has a significant role in fibrosis compared to epithelial cell-derived WISP1.

Response 2- We appreciate and thank the reviewer again for the comment. In humans, WISP1 expression has been confirmed in various tissues and the expression is cell type specific [1]. Fibroblasts (lung [2], liver [7], heart [8]), epithelial cells (lung) [9], cardiomyocytes (heart) [10], neurons [11], microglia (brains) [12], chondrocytes [13], osteoblast (bone) [14] and many more are sources of WISP1. While ATII cells are one of the sources, according to the human protein atlas, fibroblasts have the highest WISP1 expression [15]. Although fibroblasts are the primary cell responsible for ECM production leading to fibrosis, we understand that WISP1 is a secreted protein and it can elicit both autocrine and paracrine effects (ATII-derived WISP1). We addressed this as one of the limitations of our study in Discussion section from line 628-631 as “Here, we are studying primary fibroblasts in isolation, which fails to capture the paracrine effects of WISP1 and crosstalk between fibroblasts and tissue-resident cells including but not limited to epithelial cells, endothelial cells, as well as other immune cells, such as macrophages and neutrophils.”

In addition, evidence in the literature also suggests that WISP1 is upregulated in both pre-clinical (bleomycin and paraquat induced model) and clinical IPF patients [9,16-20]. For instance, 1.25U/kg and 2U/kg i.t Bleomycin administration increases WISP1 levels to 60-80pg/ml and 3-days post-bleomycin challenge the WISP1 levels reached to ~125pg/ml [19]. Similarly, WISP1 was upregulated in human whole lung tissue homogenate from 6 IPF donors by western blot with patient to patient variability [9,20]. Furthermore, as addressed in line 186-188 of results section, the variability in the endogenous expression may be due to patient to patient variability stemming from their underlying comorbidities and patient history. 

  1. The authors do not observe a direct effect of WISP1 (1000 nM) in inducing pro-fibrotic gene expression at the RNA level. Two alternative explanations should be considered: (1) WISP1’s effect on RNA might be rapid, making the 24-hour observation point too late. A 12-hour time point should be included; (2) in some cases, collagen protein levels may be more responsive than RNA levels, especially in HLF. The authors should directly measure ECM protein levels to verify that WISP1 alone does not induce fibrosis. Performing a hydroxyproline assay is recommended to assess collagen deposition directly.

Response 3- We thank the reviewer and appreciate the feedback. As suggested, to ensure that WISP1 mediated effects are time dependent (1) we looked at the effect of rD3D4-WISP1 (recombinant WISP1-domain 3 and 4) (1000nM) protein from 6-72 hrs in Normal Human Lung Fibroblast (NHLF). Our previous studies have shown that rD3D4-WISP1 acts similarly as full length (FL) WISP1 (used in the current study) in multiple biological systems, including increase in cytokines and chemokines, such as IL-6 and CCL2. Also, rD3D4-WISP1 did not show the induction of pro-fibrotic gene expression (such as COL1A1, COL3A1, ACTA2, FN1) either at an early time point such as 6hr or later at 72hr. However, IL-6 and CCL2 were significantly upregulated up until 72hrs in the same studies. As a result, we did not find it necessary to repeat these studies using WISP1 FL protein. However, given that assessing the WISP1 domain effects is beyond the scope of this paper, we will not include this data in the current manuscript, but provided here as a figure to address the reviewer's comment.

(2) We have addressed reviewers' comments in supplementary figure S5 and in line 278-279 of the main text (highlighted). To assess the effect of WISP1 FL on ECM protein levels, particularly type I collagen, we utilized immunofluorescence. Here, similar to pRB staining (Fig. 3), normal human dermal fibroblasts were plated in 96 well tissue culture treated 96 well plates at 6000 cells per well. The next day cells were serum starved for 5h then treated with various concentrations of WISP1 for 72h. Post-fixation, cells were then incubated with rabbit anti-collagen 1a1 primary antibody (Invitrogen) at 6.4ug/ml in PBS/1% BSA for 1 h, washed, with PBS then incubated with secondary Ab, goat anti Rab Cy3 (Abcam) at 1:2000 and Nuc Blue (Invitrogen) for 1h.  Cells were washed and imaged with Phenix opera utilizing Harmony software to quantify the intensity of collagen 1a1/Cy3 staining. We did not find any effect of WISP1 on collagen protein levels, similar to our RNA data, bolstering our findings. If required we are happy to provide additional data to show the effect of WISP1 on type I collagen protein through other methods.

Minor Concerns/Specific comments

  1. In Figure 1A, qPCR data: If NHLF is used as the control group for qPCR fold change, its average should be 1, not ~5 as currently shown. Please double-check the data processing, as WISP1 is generally expected to increase 4-10 fold compared to control, rather than 20-60 fold as shown in Panel A. Clearly indicate the type of error bars (S.D. or S.E.M.) in the figure legend (applies to all figures).

Response 1- We appreciate and thank the reviewer. The WISP1 expression in 5-NHLF donors had high variability potentially due to underlying conditions. Although the overall trend remain the same, i.e. NHLF had high Ct values and DHLF had low Ct values. In order to capture the patient heterogeneity and variations with respect to WISP1 expression, we normalized the data to the highest Ct value NHLF donor to also highlight patient-to-patient variability as indicated in the main text line 186-188. Data are represented as mean ± SD (main text line 173). We have also added this in the figure legends, main text line 195-196, 236, 267, 289 and 345 as requested by the reviewer. 

  1. In Figure 4, panel D lacks a label.

Response 2- We thank the reviewer and appreciate the feedback. We added a label for panel D in Figure 4. 

  1. Figures 6, 7, and 8C (pathways analysis): Clarify the meaning of the heat gradient—is it based on p-adjusted values for each pathway or the percentage of genes influenced by each condition? Additionally, I recommend listing the specific genes used in pathway/enrichment analyses for each group in Panel C (e.g., TGFB vs. Control). For Panel D, indicate which genes are highlighted as ‘TGFB+WISP1 vs TGFB’ and consider marking these genes in Panel C. Supplementary tables should clearly correspond to Panel C’s gene lists for ease of reference.

Response 3- We appreciate and thank the reviewer for the comment. To address reviewers comment and for ease of reference, we have added the list of genes used in the pathway analysis as supplementary figure S8 and cited in the main text as well in line 407-408, 482-483 and 565-566 (highlighted). For panel D, the list of genes are indicated in Supplementary Table S2 (NHDF)-cited in the main text line 386 and 392, Table S4 (NHLF)-cited in the main text line 454, 461, 464 and Table S6 (DHLF)-cited in the main text line 575, 580.

  1. For data availability, it would be beneficial if the authors could upload their raw data to GEO or, at minimum, provide the counts file as supplementary material.

Response 4- We thank the reviewer for the comment. We have provided all the statistically significant list of genes as supplementary tables S1-S8 for NHDF, NHLF and DHLF. 

References-

  1. Singh, K., & Oladipupo, S. S. (2024). An overview of CCN4 (WISP1) role in human diseases. Journal of translational medicine, 22(1), 601. https://doi.org/10.1186/s12967-024-05364-8
  2. Klee, S., Lehmann, M., Wagner, D. E., Baarsma, H. A., & Königshoff, M. (2016). WISP1 mediates IL-6-dependent proliferation in primary human lung fibroblasts. Scientific reports, 6, 20547. https://doi.org/10.1038/srep20547
  3. Jian, Y. C., Wang, J. J., Dong, S., Hu, J. W., Hu, L. J., Yang, G. M., Zheng, Y. X., & Xiong, W. J. (2014). Wnt-induced secreted protein 1/CCN4 in liver fibrosis both in vitro and in vivo. Clinical laboratory, 60(1), 29–35.https://doi.org/10.7754/clin.lab.2013.121035
  4. González, D., Campos, G., Pütter, L., Friebel, A., Holland, C. H., Holländer, L., Ghallab, A., Hobloss, Z., Myllys, M., Hoehme, S., Meindl-Beinker, N. M., Dooley, S., Marchan, R., Weiss, T. S., Hengstler, J. G., & Godoy, P. (2024). Role of WISP1 in Stellate Cell Migration and Liver Fibrosis. Cells, 13(19), 1629. https://doi.org/10.3390/cells13191629
  5. Wang, B., Ding, X., Ding, C., Tesch, G., Zheng, J., Tian, P., Ricardo, S., Shen, H. H., & Xue, W. (2020). WNT1-inducible-signaling pathway protein 1 regulates the development of kidney fibrosis through the TGF-β1 pathway. FASEB journal : official publication of the Federation of American Societies for Experimental Biology, 34(11), 14507–14520. https://doi.org/10.1096/fj.202000953R
  6. Zhang, M., Meng, Q. C., Yang, X. F., & Mu, W. D. (2020). TGF-β1/WISP1/Integrin-α interaction mediates human chondrocytes dedifferentiation. European review for medical and pharmacological sciences, 24(17), 8675–8684. https://doi.org/10.26355/eurrev_202009_22804
  7. Xi, Y., LaCanna, R., Ma, H. Y., N'Diaye, E. N., Gierke, S., Caplazi, P., Sagolla, M., Huang, Z., Lucio, L., Arlantico, A., Jeet, S., Brightbill, H., Emson, C., Wong, A., Morshead, K. B., DePianto, D. J., Roose-Girma, M., Yu, C., Tam, L., Jia, G., … Ding, N. (2022). A WISP1 antibody inhibits MRTF signaling to prevent the progression of established liver fibrosis. Cell metabolism, 34(9), 1377–1393.e8. https://doi.org/10.1016/j.cmet.2022.07.009
  8. Venkatachalam, K., Venkatesan, B., Valente, A. J., Melby, P. C., Nandish, S., Reusch, J. E., Clark, R. A., & Chandrasekar, B. (2009). WISP1, a pro-mitogenic, pro-survival factor, mediates tumor necrosis factor-alpha (TNF-alpha)-stimulated cardiac fibroblast proliferation but inhibits TNF-alpha-induced cardiomyocyte death. The Journal of biological chemistry, 284(21), 14414–14427. https://doi.org/10.1074/jbc.M809757200
  9. Königshoff, M., Kramer, M., Balsara, N., Wilhelm, J., Amarie, O. V., Jahn, A., Rose, F., Fink, L., Seeger, W., Schaefer, L., Günther, A., & Eickelberg, O. (2009). WNT1-inducible signaling protein-1 mediates pulmonary fibrosis in mice and is upregulated in humans with idiopathic pulmonary fibrosis. The Journal of clinical investigation, 119(4), 772–787. https://doi.org/10.1172/JCI33950
  10. Colston, J. T., de la Rosa, S. D., Koehler, M., Gonzales, K., Mestril, R., Freeman, G. L., Bailey, S. R., & Chandrasekar, B. (2007). Wnt-induced secreted protein-1 is a prohypertrophic and profibrotic growth factor. American journal of physiology. Heart and circulatory physiology, 293(3), H1839–H1846. https://doi.org/10.1152/ajpheart.00428.2007
  11. Wang, S., Chong, Z. Z., Shang, Y. C., & Maiese, K. (2012). WISP1 (CCN4) autoregulates its expression and nuclear trafficking of β-catenin during oxidant stress with limited effects upon neuronal autophagy. Current neurovascular research, 9(2), 91–101. https://doi.org/10.2174/156720212800410858
  12. Shang, Y. C., Chong, Z. Z., Wang, S., & Maiese, K. (2012). Wnt1 inducible signaling pathway protein 1 (WISP1) targets PRAS40 to govern β-amyloid apoptotic injury of microglia. Current neurovascular research, 9(4), 239–249. https://doi.org/10.2174/156720212803530618
  13. Hou, C. H., Chiang, Y. C., Fong, Y. C., & Tang, C. H. (2011). WISP-1 increases MMP-2 expression and cell motility in human chondrosarcoma cells. Biochemical pharmacology, 81(11), 1286–1295. https://doi.org/10.1016/j.bcp.2011.03.016
  14. Wu, C. L., Tsai, H. C., Chen, Z. W., Wu, C. M., Li, T. M., Fong, Y. C., & Tang, C. H. (2013). Ras activation mediates WISP-1-induced increases in cell motility and matrix metalloproteinase expression in human osteosarcoma. Cellular signalling, 25(12), 2812–2822. https://doi.org/10.1016/j.cellsig.2013.09.005
  15. https://www.proteinatlas.org/ENSG00000104415-CCN4/single+cell#single_cell_type_summary
  16. Liu J, Lv S, Ma W, Yang D, Zhang X. Effect of WISP1 on paraquat-induced EMT. Toxicol Vitr [Internet]. 2023;93(May):105693. Available from: https://doi.org/10.1016/j.tiv.2023.105693
  17. Sun Z, Yang Z, Wang M, Huang C, Ren Y, Zhang W, et al. Paraquat induces pulmonary fibrosis through Wnt/β-catenin signaling pathway and myofibroblast differentiation. Toxicol Lett [Internet]. 2020;333(June):170–83. Available from: https://doi.org/10.1016/j.toxlet.2020.08.004
  18. Jian, Y. C., Wang, J. J., Dong, S., Hu, J. W., Hu, L. J., Yang, G. M., Zheng, Y. X., & Xiong, W. J. (2014). Wnt-induced secreted protein 1/CCN4 in liver fibrosis both in vitro and in vivo. Clinical laboratory, 60(1), 29–35. https://doi.org/10.7754/clin.lab.2013.121035 
  19. Kadam, A. H., & Schnitzer, J. E. (2023). Characterization of acute lung injury in the bleomycin rat model. Physiological reports, 11(5), e15618. https://doi.org/10.14814/phy2.15618 
  20. 18. Berschneider, B., Ellwanger, D. C., Baarsma, H. A., Thiel, C., Shimbori, C., White, E. S., Kolb, M., Neth, P., & Königshoff, M. (2014). miR-92a regulates TGF-β1-induced WISP1 expression in pulmonary fibrosis. The international journal of biochemistry & cell biology, 53, 432–441. https://doi.org/10.1016/j.biocel.2014.06.011

Reviewer 2 Report

Comments and Suggestions for Authors

The authors provide an excellent presentation of molecular, genomic and proteomic workup of the factors influencing the initiation and progression of fibrosis. Their work is based on isolated fibroblasts, what they indicate as a main limitation of the study. Nevertheless, they plan to continue their work and use more advanced cellular systems, which may increase the reliability of their findings. As the treatments of pulmonary fibrosis are still unsatisfactory and we desperately need new active particles in my opinion any research that shows interactions between different molecules and fibrosis is important. I have only small remarks regarding the clinical part of the manuscript:

1.       Pirfenidone and nintedanib do not provide the symptomatic relief but they slow the disease progression

2.       The side effects of antifibrotic drugs rarely are severe, usually they wane after few months and they are mostly menagable

Author Response

Reviewer #2-

  1.       Pirfenidone and nintedanib do not provide the symptomatic relief but they slow the disease progression

Response 1- We thank the reviewer for the comment. We have addressed it in the main text line 15 and 48, highlighted.

  1.       The side effects of antifibrotic drugs rarely are severe, usually they wane after few months and they are mostly manageable

Response 2- We appreciate and thank the reviewer for the comment. We rephrased the sentence and removed the word ‘severe’ from line 49 and addressed it in the main text from line 50-51, highlighted.

Round 2

Reviewer 1 Report

Comments and Suggestions for Authors

The authors have addressed all my concerns. The revised manuscript is suitable for publication after proofreading by the authors.